# Assessing the inter- & intra-reliability of a customised volleyball performance analysis system to analyse complexes and the efficacy of the associated skills

**Steven Nicklin**[ID]*, **Lee Nelson, Evelyn Carnegie, Jayamini Ranaweera, Greg Doncaster**

Department of Sport & Physical Activity, Edge Hill University, Ormskirk, United Kingdom

* Steven.Nicklin@edgehill.ac.uk

## Abstract

Within volleyball, performance analysis is employed to identify key performance indicators (KPIs) in relation to success. To date, however, despite the growth in performance analysis, there is little appreciation of the assessment, importance and reporting of reliability of individual variables. This study seeks to establish the inter- and intra-reliability of a bespoke notational analysis system designed for assessing volleyball performance, considering complexes, position, skill, type of skill, and skill efficacy. To establish the inter-reliability, two coders analysed 9 matches, randomly selected from a pool of elite international matches. In addition, to establish the intra-reliability, the primary coder re-analysed one match per month over a nine-month data collection period. Cohen's Kappa ($\kappa$) were established for nominal data, whilst the values for skill efficacy were reported as a Weighted Kappa alongside an Inter-Class Correlation (ICC). Kappa and Weighted Kappa values for all data were shown to be above the acceptable thresholds ($\kappa = 0.75$, $\kappa = 0.60$, respectively). The present study demonstrates that it is plausible for a bespoke volleyball coding system, implementing a wide array of variables, to have high levels of inter- and intra-reliability. Findings highlight that it is possible for such a system to be utilised over a prolonged period of data collection.

## Introduction

Within sports performance, notational analysis is employed as an objective method to identify key performance indicators (KPIs) or areas for improvement. The data collected can inform practice by aiding and informing feedback, assisting in talent identification and development processes, and developing training procedures [1]. However, for the data to be used for such purposes, it must demonstrate appropriate levels of reliability and validity [2]. O'Donoghue [3] defines reliability as the consistency of values recorded when a measurement is used. Whilst reliability can be

**Data availability statement:** \*\*\*A/PROS AT ACCEPT: At FTC please follow up with authors to see that data is accessible in the provided repository and update DAS\*\*\*Upon review and further discussion, we will make the data from this study available through the institutional repository (PURE) (https://doi.org/10.25416/edgehill.30598397). However, they will only permit it to be uploaded upon confirmation of paper acceptance. We sought reasonable request for access to the data set due to its position within a larger research project.

**Funding:** The author(s) received no specific funding for this work.

**Competing interests:** The authors have declared that no competing interests exist.

assessed in various methods, the most commonly implemented methods within the performance analysis literature are inter-rater reliability and test-retest (intra-rater) reliability [3,4].

Inter-rater reliability refers to the reliability of the coding instrument, ensuring that it can be used accurately by multiple analysts to identify the variables in question [3]. However, due to the number of coders used, even the most clearly defined operational definitions can still be subject to an individual's interpretation (individuals' "subjectivity") [5]. As such, inter-rater reliability requires clearly defined and objective operational definitions, thus permitting numerous coders to correctly and consistently identify, define and agree on the performance analysis procedures [3]. Ultimately, high levels of inter-reliability are necessary as this indicates that the coding system can effectively be used by multiple coders [6]. Intra-reliability, however, seeks to assess the reliability of an individual coder to consistently identify data points multiple times when undertaking the same analysis [7,8]. As this method of assessment requires a coder to re-analyse a previously evaluated performance, there is the potential for the "learning effect" to occur [3]. Therefore, an appropriate amount of time should be afforded between instances of repeat coding. Yet, within the literature, this varies from seven [9] to fourteen days [3], with some failing to report a break at all [10]. The process of re-analysing also aids in the identification of potential observer drift, when the coder's interpretation and application of the operational definitions (potentially) changes during a prolonged period of coding, something that is particularly pertinent to those working in practise (e.g., throughout a competitive season) [11].

Research into performance analysis of volleyball has typically considered the successful and unsuccessful performance of skills and the corresponding types [12–14], which has led to the reliability of such processes being widely reported and established. In response to calls within the literature, recent research has sought to investigate volleyball performance at a granular level by including efficacy scales and complexes [15–17]. However, research has explored the reliability of efficacy scales [18] and complexes [19] in isolation, reporting Kappa values of >.81 and .913-1.000 respectively. Notably, there is no evidence of studies considering the inter- and intra-reliability of a system that incorporates complexes, position, skill, type of skill, and skill efficacy, despite there being further calls within the literature to integrate both complexes and efficacy scales into the analysis of volleyball performance [19–21]. Therefore, future research should consider the assessment of inter- and intra-reliability, and the implementation of such a detailed coding system for analysing volleyball performance.

Despite the known importance of the assessment of reliability, throughout the existing volleyball literature there is varied practise in the consideration, analysis, and reporting of (inter and intra-reliability. Indeed, there is evidence of authors opting to present Kappa values as a range [14,22,23] or by claiming they exceed a given value or threshold [20,24,25]. Deciding to present the values in such methods prevents future research from comparing and scrutinising the reliability of individual variables. As such, future research should seek to be more transparent in its reporting of

inter- and intra-reliability by presenting the kappa values of individual variables. Furthermore, despite intra-reliability providing an opportunity to identify observer drift, studies tend to only re-analyse one match from their sample once [26,27] rather than multiple instances across a longer period of time. Therefore, leaving the potential impact of observer on the data collected as unknown. To address this issue, future research into notational analysis systems across all sports should seek to conduct the assessment of intra-reliability over a prolonged period of time to ensure that observer drift has not taken affect and present all the kappa values from this process to aid with the transparency.

Therefore, the purpose of the current study was to examine the inter- and intra-reliability of a bespoke volleyball notational analysis system designed to record complexes, player position, skill, type of skill, and skill efficacy. In addition, the current study will also seek to provide the findings as stratified values for each variable assessed. Finally, taking into account the potential impact of observer drift, the current study will conduct the assessment of intra-reliability over a prolonged period of time. An improved understanding of inter- and intra-reliability for each of these distinct categories will equip performance analysts and coaches with a reliable system for detailed performance analysis, facilitating talent development, identification, and targeted training practices.

## Methods

The process of establishing both inter- and intra-reliability for this study consisted of five phases: the construction of the coding window, the recruitment and training of the secondary coder, the assessment of inter-reliability, the assessment of intra-reliability, and the analysis of the data.

### Construction of the coding window

The bespoke coding system used in this study was designed using Nacsport Scout+ (Nacsport 6.5, Nacsport, Spain) to permit the detailed recording of each action. The process of coding individual actions followed the same pattern to allow for a simpler analysis process. The initial step in coding was to assign the relevant Set button using the auto-assign descriptor function in Nacsport Scout+; this was altered between sets to allow for distinction in future analysis. The model of complexes used in this study, and the corresponding definitions, were adopted from Laporta et al. [20]. Following this, the relevant team and player position were identified based upon the P2 forms that the Fédération Internationale de Volleyball (FIVB) publish after each match. In accordance with Rebelo et al. [28], serve, serve receive, set, spike, block, and defence were included in the coding system. Next, each of these key skills were assigned a five-point efficacy scale, with individualised operational definitions, informed by the literature, to allow for an easier and more precise coding process. The final step was to identify the action's effect on the rally (Point Won, Continuation, or Point Lost).

### Recruitment and training

Following the construction of the coding window, a secondary coder was required to establish the inter-reliability of the coding system and the associated operational definitions. Secondary coder inclusion criteria required them to have volleyball experience both as a coach and an athlete at a high level, and still active in the sport. The primary and secondary coders had a wide range of high level national and international volleyball experience across at least 10 years as coaches and athletes.

Once the secondary coder was made aware of what the role entailed as part of this study, the primary researcher began the training process. This consisted of one session per week over a four-week period, in which the five-step model adapted from Painczyk, Hendricks, and Kraak [10] was implemented. The process began with a familiarisation of the operational definitions for complexes (Table 1), and then the efficacy scales for each skill (Table 2), using an isolated rally not associated with the sample. Following the familiarisation, the second coder was introduced to Nacsport Scout+ and the coding window, including how to start and end a recording, alter the auto-descriptors, and edit the data captured whilst observing the footage. The second coder was then permitted to code fifteen points of a random match under the supervision of the primary coder, with input and support provided when required. The final step of the training process entailed

**Table 1. The operational definitions and skills involved in complexes, and the phase of a rally they refer to.**

| Complex (K) | Phase of the match | Skills involved | Definitions |
|---|---|---|---|
| K0 | Service | Serve | This complex begins each rally and contains only one skill. It is often considered to be an isolated complex as it isn't dependent on a previous action. |
| K1 | Serve Receive | Serve Receive, Set, Spike | Within this complex a team will look to receive the serve from the opposition in order to create a good attacking opportunity. |
| K2 | Block Defence | Block, Defence, Set, Spike | This is the first phase of play for the serving team. They will look to block the opposition attack before organising an attack of their own. |
| K3 | Open Play | Block, Defence, Set, Spike | This complex often feeds into itself with both teams attacking and defending. |
| K4 | Attack Coverage | Defence, Set, Spike | This complex occurs following a successful block by the opposition, forcing the attacking team to defend the block then attack. |
| K5 | Freeball | Block, Defence, Set, Spike | This complex occurs when the opposition are unable to perform an attack and are forced to pass the ball over the net. |

**Table 2. The operational definitions for the efficacy scales of terminal and continuous volleyball skills, and the respective technique.**

| Skill | 0 | 1 | 2 | 3 | 4 | Types of Skill |
|---|---|---|---|---|---|---|
| *Serve* | Error (Point Lost) | Easy receive for the opponent, all attacking options available | Received by opponents, limited attacking options | Difficult receive, out of system attack or freeball | Ace (Point Won) | *Power (Jump and Standing)* *Float (Jump and Standing)* |
| *Spike* | Error (Point Lost) | Easily Defended by Opposition | Spike attempted but was dug up by the opposition with limited attacking options | Spike attempted but was dug up by the opposition with difficulty | Kill (Point Won) | *Diagonal* *Line* *Swipe* *Tip* |
| *Block* | Unforced Errors such as net touch or centre line fault (Point Lost) | Block that the opponent hits through or directs it away from the defence (Forced Errors). | Block that allows the rally to continue without an advantage (for either side) | Block that forces the opposition to defend (back to the attacking side) | Stuff Block (Point Won) | *1 Person* *2 Person* *3 Person* |
| *Serve Receive* | Error (Point Lost) | Ball sent directly back to the server or is passed away from the setter | Ball can be set, limited attacking options or out-of-system attack | Ball can be set, but no quick attacks are available | Ball can be set with all attacking options (Perfect Receive) | *Overhead* *Underarm* |
| *Set* | Error (Point Lost) | Set in the general area of the hitter but with limited hitting options | Set in a good area but the hitter has limited attacking options | Set in a good area and the hitter has all attacking options | Set in the best area for the hitter with all attacking options (Perfect Set) | *Tempo 1* *Tempo 2* *Tempo 3* *2nd Contact Attack* |
| *Defence* | Error (Point Lost) | Ball sent directly back to the server or is passed away from the setter | Ball can be set, limited attacking options or out-of-system attack | Ball can be set, but no quick attacks are available | Ball can be set with all attacking options (Perfect Pass) | *Overhead* *Underarm* |

the second coder coding a match without any external support. This gradually built up from a single set until the second coder was comfortable in coding a complete match without support. Before analysing any footage as part of the present study, the secondary coder conducted thirteen hours of training over the four-week training period (3.25 hours per week), to ensure optimal understanding of the coding window and was confident in applying and identifying the relevant operational definitions for complexes, player position, skill, type of skill (S1-S6), and skill efficacy.

## Inter- and intra-reliability assessment

Upon the completion of the training process, both the primary researcher and secondary coder undertook the inter-reliability process separately, in which both observers were required to record complexes, player position, skill, type of

skill, and skill efficacy, for each contact with the ball, using the bespoke coding window. Both coders were permitted twelve weeks (two sessions per week) to code nine volleyball matches (equating to 8082 ball contacts and 40410 data points, per coder). This accounted for 13% of the total number of matches available from the entire pool of available matches (n = 68), from elite level male world championships.

Parallel to the establishment of inter-reliability, the intra-reliability (for the primary coder) commenced at the same time but then continued over a longer nine-month data collection period. Furthermore, in consideration of observer drift, intra-reliability was considered throughout the entirety of the prolonged period of data collection (Fig 1). Specifically, this comprised of 9 four-week blocks, in which each block consisted of an initial period of 3 weeks of data collection, then a minimum of a 1-week break from coding, after which one game from that respective block was reanalysed for intra-reliability purposes. This resulted in a total of nine matches (equating to 6906 ball contacts and 34530 data points) being reanalysed to establish intra-reliability.

## Statistical analysis

For both the inter- and intra-reliability, the recorded data was initially exported from Nacsport Scout+ to Microsoft Excel version 2410 (Microsoft 365 2024, Microsoft, Redmond, Washington), where it was cleaned to ensure that the start points of rallies were aligned across coders/instances for each point (i.e., both start at K0 using the serve). In line with the procedures outlined by O'Donoghue [3] and De Raadt et al. [29], the data collected was then processed and cleaned; instances of missing data resulted in the deletion of all data in that line from the data set. The (cleaned) data was then transferred to SPSS version 25.0.0.1 (IBM SPSS Statistics for Windows, IBM, Armonk, New York), where a Cohen's Kappa was calculated for nominal data (e.g., complexes, position, skill, and type of skill), with a Weighted Cohen's Kappa was calculated for ordinal data (e.g., skill efficacy). To aid interpretation within the literature, thresholds for kappa values were employed, in which 0.81–0.99 = *Almost Perfect Agreement*; 0.61–0.80 = *Substantial Agreement*; 0.41–0.60 = *Moderate Agreement*; 0.21–0.40 = *Fair Agreement*; 0.01–0.20 = *Slight Agreement*; and < 0 = *Less Than Chance Agreement* [30,31]. As noted by O'Donoghue [3], Weighted Kappas provide a greater appreciation of the slight discrepancies in ordinal data, resulting in a reduced threshold (κ = 0.60) being proposed to accommodate such differences [32]. In addition, to further assess the reliability, an Inter-Class Correlation (ICC) was conducted for skill efficacy in which the thresholds of >0.90 = Excellent Reliability, 0.75–0.90 = Good Reliability, 0.50–0.75 = Moderate Reliability, <0.50 = Poor Reliability [33], were considered. Finally, for each block and overall values, 95% confidence intervals were calculated for Kappa, Weighted Kappa, and ICC, for both inter- and intra-reliability.

## Results

### Inter-reliability

Table 3 shows the Kappa and ICC values for the individual matches. Regarding the variability of variables across the nine matches, discrepancies were reported, with the largest being reported for Type of Set (κ = 0.33–0.62). Table 4 provides the Kappa and ICC values, the confidence intervals for three blocks of matches, and the overall data set, along with the

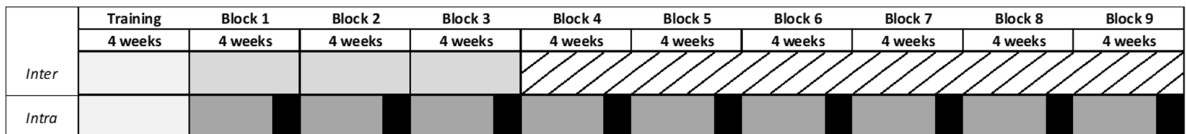

**Fig 1. A timeline demonstrating the data collection processes for assessing the intra-reliability of the bespoke coding window, with consideration for observer drift.**

**Table 3. Inter-reliability results represented as Kappa values and Confidence Intervals for individual matches.**

| Performance Indicators | Block 1 | | | Block 2 | | | Block 3 | | |
|---|---|---|---|---|---|---|---|---|---|
| | Match 1 | Match 2 | Match 3 | Match 4 | Match 5 | Match 6 | Match 7 | Match 8 | Match 9 |
| **Complex** | 0.95 | 0.96 | 0.97 | 0.95 | 0.96 | 0.96 | 0.95 | 0.99 | 0.98 |
| **Player Position** | 0.89 | 0.89 | 0.92 | 0.86 | 0.92 | 0.93 | 0.92 | 0.92 | 0.90 |
| **Skill** | 0.98 | 0.97 | 0.98 | 0.98 | 0.99 | 0.97 | 0.99 | 0.97 | 0.98 |
| **Skill Type** | 0.80 | 0.75 | 0.77 | 0.73 | 0.79 | 0.75 | 0.74 | 0.79 | 0.76 |
| *Type of Block* | 0.91 | 0.89 | 0.95 | 0.83 | 0.88 | 0.89 | 0.94 | 0.91 | 0.90 |
| *Type of Defence* | 0.93 | 1.00 | 0.94 | 0.93 | 0.96 | 0.93 | 0.90 | 0.95 | 0.97 |
| *Type of Serve* | 0.87 | 0.88 | 0.82 | 0.65 | 0.85 | 0.69 | 0.69 | 0.84 | 0.73 |
| *Type of Serve Receive* | 0.98 | 0.97 | 0.98 | 1.00 | 1.00 | 0.99 | 1.00 | 0.98 | 0.98 |
| *Type of Set* | 0.60 | 0.44 | 0.45 | 0.33 | 0.49 | 0.62 | 0.51 | 0.57 | 0.47 |
| *Type of Spike* | 0.53 | 0.38 | 0.49 | 0.55 | 0.52 | 0.47 | 0.55 | 0.58 | 0.45 |
| **Efficacy** | 0.63 | 0.65 | 0.63 | 0.79 | 0.59 | 0.67 | 0.70 | 0.76 | 0.59 |
| **Inter-Class Correlation** | | | | | | | | | |
| Performance Indicators | Block 1 | | | Block 2 | | | Block 3 | | |
| | Match 1 | Match 2 | Match 3 | Match 4 | Match 5 | Match 6 | Match 7 | Match 8 | Match 9 |
| **Efficacy** | 0.88 | 0.91 | 0.89 | 0.95 | 0.95 | 0.92 | 0.91 | 0.94 | 0.88 |

**Table 4. Inter-reliability results represented as Kappa values and Confidence Intervals for the blocks of matches.**

| Performance Indicators | Block 1 | Block 2 | Block 3 | Overall Agreement Interpretation | |
|---|---|---|---|---|---|
| | Average (95% CI) | Average (95% CI) | Average (95% CI) | Average (95% CI) | Interpretation |
| **Complex** | 0.96 (0.95 - 0.97) | 0.95 (0.95 - 0.96) | 0.97 (0.97 - 0.98) | 0.96 (0.96 - 0.97) | *Almost Perfect* |
| **Player Position** | 0.90 (0.89 - 0.91) | 0.91 (0.90 - 0.92) | 0.91 (0.90 - 0.93) | 0.91 (0.90 - 0.92) | *Almost Perfect* |
| **Skill** | 0.98 (0.98 - 0.99) | 0.98 (0.97 - 0.98) | 0.98 (0.97 - 0.98) | 0.98 (0.97 - 0.98) | *Almost Perfect* |
| **Skill Type** | 0.78 (0.76 - 0.79) | 0.76 (0.75 - 0.78) | 0.76 (0.74 - 0.76) | 0.77 (0.76 - 0.78) | *Substantial Agreement* |
| *Type of Block* | 0.92 (0.88 - 0.95) | 0.88 (0.85 - 0.92) | 0.91 (0.88 - 0.94) | 0.90 (0.89 - 0.92) | *Almost Perfect* |
| *Type of Defence* | 0.95 (0.93 - 0.97) | 0.95 (0.93 - 0.97) | 0.94 (0.91 - 0.96) | 0.95 (0.93 - 0.96) | *Almost Perfect* |
| *Type of Serve* | 0.86 (0.81 - 0.90) | 0.78 (0.74 - 0.82) | 0.75 (0.69 - 0.79) | 0.79 (0.76 - 0.82) | *Substantial Agreement* |
| *Type of Serve Receive* | 0.98 (0.96 - 0.99) | 0.99 (0.99 - 1.00) | 0.99 (0.97 - 1.00) | 0.99 (0.98 - 0.99) | *Almost Perfect* |
| *Type of Set* | 0.50 (0.45 - 0.55) | 0.52 (0.47 - 0.57) | 0.50 (0.45 - 0.56) | 0.51 (0.48 - 0.54) | *Moderate Agreement* |
| *Type of Spike* | 0.47 (0.42 - 0.53) | 0.50 (0.45 - 0.56) | 0.51 (0.45 - 0.56) | 0.49 (0.46 - 0.52) | *Moderate Agreement* |
| **Efficacy** | 0.63 (0.61 - 0.65) | 0.66 (0.64 - 0.68) | 0.67 (0.65 - 0.68) | 0.65 (0.64 - 0.67) | *Substantial Agreement* |
| **Inter-Class Correlation** | | | | | |
| Performance Indicators | Block 1 | Block 2 | Block 3 | Overall Agreement Interpretation | |
| | Average (95% CI) | Average (95% CI) | Average (95% CI) | Average (95% CI) | Interpretation |
| **Efficacy** | 0.89 (0.88 - 0.90) | 0.90 (0.89 - 0.91) | 0.91 (0.90 - 0.91) | 0.90 (0.89 - 0.90) | *Excellent Reliability* |

interpretation of values. The variability reported across the nine matches is reduced across the three blocks compared with the values across the nine weeks, with the largest discrepancy being reported for Type of Spike (κ = 0.47–0.51). In relation to the overall inter-reliability Complex, Position, Skill, Skill Type, Type of Serve, Type of Serve Receive, Type of Block, and Type of Defence produced Kappa values above the relevant threshold (κ = 0.75). However, Type of Set and Type of Spike displayed a greater variance between coders, producing Kappa values of κ = 0.51 and κ = 0.49, respectively. Additionally, Efficacy produced a Kappa value above the Weighted Kappa threshold (κ = 0.60) and an ICC value of 0.90, reaffirming the reliability of this variable.

## Intra-reliability

Table 5 presents the Kappa and ICC values for individual matches in the intra-reliability sample. The variability reported across the nine matches for intra-reliability highlights a reduction in discrepancies across all variables, apart from Type of Serve (κ=0.48–0.96). Table 6 provides the Kappa and ICC values and the confidence intervals for the three blocks, with agreement interpretations also included for the overall kappa values. Similarly to the inter-reliability values for the three blocks, the variation reported is minimal except Type of Serve (κ=0.72–0.97). With regards to the overall intra-reliability Complex, Position, Skill, Skill Type, Type of Serve, Type of Serve Receive, Type of Block, and Type of Defence produced a Kappa value above the acceptable threshold (κ=0.75). However, Type of Set and Type of Spike displayed a greater variance between coding instances, reporting Kappa values of κ=0.70 and κ=0.73, respectively, which, whilst demonstrating *Substantial Agreement,* did not exceed the threshold. In addition, Efficacy produced a Kappa value above the Weighted Kappa threshold (κ=0.60) and an ICC value of 0.94, reaffirming the reliability of the variable.

## Discussion

The aim of this study was to establish the inter- and intra-reliability of a bespoke coding window for analysing volleyball performance. The main findings are that the development of a bespoke coding window was shown to have acceptable levels of inter- and intra-reliability and is therefore appropriate for use in analysing competitive volleyball match-play. Furthermore, the rigorous (40-week) approach to assessing intra-reliability provided no indication of observer drift, therefore demonstrating that it is possible for an observer to maintain their interpretation and implementation of a bespoke coding window over a prolonged period.

Previously, literature which has analysed five or six variables has tended to report kappa values to cover all variables, in which inter-reliability kappa values range between 0.83–1. [13,14,22,23,34–36] and intra-reliability kappa values range between 0.82–1.00 [13,22,23,34–38]. In contrast, the present study reports kappa values for individual variables, which align with what has been previously shown within the literature for a similar number of variables. Specifically, the inter- and intra-reliability ranges reported by Sanchez-Moreno et al. [22] and Silva et al. [23] for analysing the characteristics of the skill setting (0.84–0.91; 0.82–0.92) and the influences of the rotation on match outcome (0.98–1.00; 0.99–1.00) within elite

**Table 5. Intra-reliability results represented as Kappa values and Confidence Intervals for individual matches.**

| Performance Indicators | Block 1 | | | Block 2 | | | Block 3 | | |
|---|---|---|---|---|---|---|---|---|---|
| | Match 1 | Match 2 | Match 3 | Match 4 | Match 5 | Match 6 | Match 7 | Match 8 | Match 9 |
| **Complex** | 1.00 | 1.00 | 0.99 | 0.99 | 1.00 | 1.00 | 1.00 | 0.99 | 1.00 |
| **Player Position** | 0.96 | 0.97 | 0.96 | 0.95 | 0.97 | 0.98 | 0.97 | 0.96 | 0.97 |
| **Skill** | 1.00 | 1.00 | 0.99 | 0.99 | 0.99 | 1.00 | 0.99 | 0.99 | 1.00 |
| **Skill Type** | 0.85 | 0.85 | 0.88 | 0.86 | 0.88 | 0.90 | 0.89 | 0.87 | 0.86 |
| *Type of Block* | *0.91* | *0.91* | *0.97* | *0.93* | *0.95* | *0.96* | *0.95* | *0.99* | *0.88* |
| *Type of Defence* | *0.99* | *0.99* | *0.99* | *0.99* | *0.99* | *0.98* | *0.95* | *0.99* | *1.00* |
| *Type of Serve* | *0.92* | *0.48* | *0.66* | *0.81* | *0.94* | *0.96* | *0.82* | *0.90* | *0.67* |
| *Type of Serve Receive* | *0.99* | *1.00* | *1.00* | *1.00* | *0.97* | *0.97* | *0.97* | *1.00* | *1.00* |
| *Type of Set* | *0.69* | *0.69* | *0.71* | *0.65* | *0.64* | *0.80* | *0.71* | *0.61* | *0.75* |
| *Type of Spike* | *0.65* | *0.72* | *0.75* | *0.74* | *0.78* | *0.70* | *0.79* | *0.66* | *0.81* |
| **Efficacy** | 0.74 | 0.82 | 0.80 | 0.81 | 0.80 | 0.80 | 0.82 | 0.74 | 0.77 |
| **Inter-Class Correlation** | | | | | | | | | |
| Performance Indicators | Block 1 | | | Block 2 | | | Block 3 | | |
| | Match 1 | Match 2 | Match 3 | Match 4 | Match 5 | Match 6 | Match 7 | Match 8 | Match 9 |
| **Efficacy** | 0.93 | 0.96 | 0.95 | 0.95 | 0.95 | 0.95 | 0.95 | 0.93 | 0.94 |

Table 6. Intra-reliability results represented Kappa values and Confidence Intervals for the three-match blocks.

| Performance Indicators | Block 1 | Block 2 | Block 3 | Overall Agreement Interpretation | |
|---|---|---|---|---|---|
| | Average (95% CI) | Average (95% CI) | Average (95% CI) | Average (95% CI) | Interpretation |
| Complex | 1.00 (0.99 - 1.00) | 0.99 (0.99 - 1.00) | 0.99 (0.99 - 1.00) | 1.00 (0.99 - 1.00) | Almost Perfect Agreement |
| Player Position | 0.96 (0.95 - 0.97) | 0.97 (0.96 - 0.98) | 0.97 (0.96 - 0.98) | 0.97 (0.96 - 0.97) | Almost Perfect Agreement |
| Skill | 0.99 (0.99 - 1.00) | 0.99 (0.99 - 1.00) | 0.99 (0.99 - 1.00) | 0.99 (0.99 - 1.00) | Almost Perfect Agreement |
| Skill Type | 0.87 (0.85 - 0.88) | 0.88 (0.87 - 0.90) | 0.87 (0.86 - 0.88) | 0.87 (0.86 - 0.88) | Almost Perfect Agreement |
| Type of Block | 0.93 (0.90 - 0.96) | 0.95 (0.92 - 0.97) | 0.94 (0.92 - 0.97) | 0.94 (0.93 - 0.96) | Almost Perfect Agreement |
| Type of Defence | 0.99 (0.98 - 1.00) | 0.99 (0.97 - 1.00) | 0.98 (0.97 - 0.99) | 0.99 (0.98 - 0.99) | Almost Perfect Agreement |
| Type of Serve | 0.72 (0.66 - 0.79) | 0.91 (0.87 - 0.94) | 0.81 (0.76 - 0.85) | 0.82 (0.79 - 0.85) | Almost Perfect Agreement |
| Type of Serve Receive | 1.00 (0.99 - 1.00) | 0.97 (0.95 - 0.99) | 0.99 (0.98 - 1.00) | 0.99 (0.98 - 0.99) | Almost Perfect Agreement |
| Type of Set | 0.71 (0.65 - 0.76) | 0.69 (0.63 - 0.75) | 0.69 (0.64 - 0.74) | 0.70 (0.67 - 0.73) | Substantial Agreement |
| Type of Spike | 0.71 (0.66 - 0.77) | 0.74 (0.68 - 0.79) | 0.75 (0.69 - 0.79) | 0.73 (0.70 - 0.76) | Substantial Agreement |
| Efficacy | 0.79 (0.77 - 0.80) | 0.81 (0.79 - 0.82) | 0.77 (0.76 - 0.79) | 0.79 (0.78 - 0.80) | Substantial Agreement |
| Inter-Class Correlation | | | | | |
| Performance Indicators | Block 1 | Block 2 | Block 3 | Overall Agreement Interpretation | |
| | Average (95% CI) | Average (95% CI) | Average (95% CI) | Average (95% CI) | Interpretation |
| Efficacy | 0.94 (0.94 - 0.95) | 0.95 (0.95 - 0.95) | 0.94 (0.93 - 0.94) | 0.94 (0.94 - 0.95) | Excellent Reliability |

male volleyball, highlight that it is possible to achieve high reliability when performing detailed analysis on one aspect of performance. A limitation of the previous research, however, is that authors tend not to report the kappa values of individual variables, unlike the present study (Tables 3–6). The presentation of findings as stratified values rather than a range provides an enhanced understanding of the reliability of individual variables and permits the comparison of values with future research.

The findings of this study highlights that the coding window proposed can reliably be used to identify for the identification of complexes, player positions, skills, and the associated skill efficacy (Tables 4 and 6). However, Skill Type requires further research and investigation as it was evident in the current data set that "subjectivity" due to the number of categories, the clarity of operational definitions, and footage angles impacted the reliability. Here, some of the challenges when undertaking the coding were that the side-on camera angle employed in volleyball can make the identification of setting tempos difficult. Similarly, when a hitter spikes the ball into a block from a short distance, the camera angle often requires the coder to guess what the intended shot was based on body positioning. Furthermore, despite being clearly defined, the interpretation of the operational definitions is still open to the "subjectivity" of coders, causing a decrease in reliability. As such, when defining and coding volleyball match-play to the level of skills type, greater consideration towards the consistent definition of skill types in volleyball, given their complexity, is needed. However, future research should seek to analyse whether it is possible to reliably code to this level (i.e., skill types), once revisions have been made to reduce elements of subjectivity.

The present study has sought to provide a greater appreciation of the categories for variables, a process that contrasts with the existing literature, by presenting the kappa values of individual types of skill. Research investigating types of skills in volleyball has tended to report a single kappa value, representing a threshold that a collection of skill types exceeds [14,39]. Building upon existing literature, current results provide a more granular assessment of the inter- and intra-reliability of types of skills, via the provision of individual kappa values for each type of skill. In the present study, the intra-reliability kappa scores for all skill types (≥ 0.70) were greater than those for inter-reliability (≥ 0.49), with types of block, defence, and serve receive reporting higher scores than those reported for types of serve, set, and spike. Nevertheless, this does raise broader questions about the clarity of operational definitions used when providing a more

detailed analysis of volleyball match-play (i.e., serve, set, and spike types), especially when interpreted by multiple coders (inter-reliability). Indeed, Williams [40] and Hughes [41] remind us that low levels of reliability often stem from issues relating to the clarity and understanding of operational definitions as well as the consistency with which these are applied when analysing performance. The variation of inter-reliability kappa values (0.48–0.96) for the type of serve could be due to multiple reasons, including the inherent subjectivity of identifying types of skills, the context of the match, or the technique used by individuals being better able to disguise the type of serve used, which would be beyond the scope of the current coding window. Therefore, moving forward, future research should seek to clearly define the operational definitions of a coding window in collaboration with expert practitioners in the sport. In addition, future research should also consider investigating the reasons for potential variation in coding variables, especially when considering reliability assessments. As within the present study, analysts and researchers should seek to employ an appropriate period of training to aid the understanding and clarity of the operational definitions before the application within a coding system.

Observer drift is an underappreciated element of performance analysis that the existing literature fails to address. The current study sought to address this within the assessment and implementation of the intra-reliability protocol. Indeed, Van der Mars [42] noted that establishing reliability in the initial stages of the use of a coding system does not ensure it will remain of an acceptable standard in perpetuity (i.e., observer drift). Despite calls for its consideration in the sporting literature [43–46], observer drift has received limited attention and is an under-investigated phenomenon within the notational analysis of sporting performance, especially in volleyball. Cobb et al. [47] investigated the potential effects of observer drift on the implementation of an athlete behaviour observation system, one week and four weeks after the initial coding instances. A drift in the percentage agreements and Yule's Q values was reported for both retests, demonstrating that observer drift occurred even over a short period [47]. In the current study, the results showed minimal variation across the rigorous, nine-month data collection period, and therefore, there was no effect of observer drift. To the best of the author's knowledge, this is the first study to investigate observer drift for a complex notational analysis system with a focus on sporting performance rather than coach or athlete behaviours [46]. Whilst the present study demonstrates that it is possible to implement a complex coding window without observer drift affecting the data collected, processes should still be undertaken, in both research and practice, to ensure that there is no drift when coding over a prolonged period (i.e., a competitive season). Therefore, future research should seek to establish appropriate procedures which can be applied consistently to research and practice, irrespective of the complexity of the coding system, to assess and monitor the potential effect of observer drift on those utilising the system.

The current study provides a robust method for the analysis of inter- and intra-reliability, and for the monitoring of observer drift; however, there are some limitations. Notably, only one additional coder was recruited to aid in the establishment of inter-reliability, whereas previous literature has sought to use multiple coders. Given the context in which this study was conducted, this was the most appropriate method to assess the inter-reliability. However, the use of only one secondary coder limits the applicability of these findings to a broader population (i.e., situations when multiple coders are employed). Therefore, future research should seek to include multiple secondary coders to improve the generalisability of such research. Furthermore, given the lack of research and procedures regarding the assessment of observer drift, the process undertaken can be regarded as rigorous and novel; however, to further understand the potential effects of observer drift, the study could have reanalysed the first match in the data set to monitor any potential drift following nine months of coding.

While there is little evidence of empirical research being conducted on the reliability of detailed notational analysis systems, the present system seeks to address this with a bespoke coding system for volleyball that incorporates both complexes and efficacy scales. It shows that it is plausible that a detailed, bespoke coding system can be considered as having high levels of inter- and intra-reliability. Furthermore, the study demonstrates that it is possible to use such a system over a prolonged period without encountering the effects of observer drift. Despite this, the study emphasises the need for future research to establish optimal procedures for the assessment and monitoring of observer drift in practice

and academia. In addition, future research should seek to utilise the current coding system to gain an improved under-standing of the extent to which key skills, within complexes, impact upon successful performance, with consideration to in- and out- of system match-play. Also, future research should give greater appreciation to the nuanced complexity of volleyball performance when defining the efficacy scales for individual skills. Finally, acknowledgment of the appreciation of the sequential nature of volleyball match-play, including the skills and associated outcomes, is warranted.

## Supporting information

**S1 Table. Definitions for Serve and Type of Serve.**
(DOCX)

**S2 Table. Definitions of Spike and Type of Spike.**
(DOCX)

**S3 Table. Definitions of Block and Type of Block.**
(DOCX)

**S4 Table. Definitions of Serve Receive and Type of Serve Receive.**
(DOCX)

**S5 Table. Definitions of Set and Type of Set.**
(DOCX)

**S6 Table. Definitions of Defence and Type of Defence.**
(DOCX)

## Author contributions

**Conceptualization:** Steven Nicklin, Lee Nelson, Evelyn Carnegie, Greg Doncaster.

**Data curation:** Steven Nicklin.

**Formal analysis:** Steven Nicklin, Lee Nelson, Jayamini Ranaweera.

**Investigation:** Steven Nicklin.

**Methodology:** Steven Nicklin, Lee Nelson, Jayamini Ranaweera.

**Project administration:** Steven Nicklin.

**Resources:** Steven Nicklin.

**Supervision:** Greg Doncaster.

**Visualization:** Greg Doncaster.

**Writing – original draft:** Steven Nicklin.

**Writing – review & editing:** Steven Nicklin, Lee Nelson, Evelyn Carnegie, Greg Doncaster.

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
