## [Decision Letter · Decision Letter 0]

6 Oct 2025

Dear Dr. Nicklin,

Thank you for submitting your manuscript to PLOS ONE. After careful consideration, we feel that it has merit but does not fully meet PLOS ONE’s publication criteria as it currently stands. Therefore, we invite you to submit a revised version of the manuscript that addresses the points raised during the review process.

**ACADEMIC EDITOR:**

We look forward to receiving your revised manuscript.

Kind regards,

Gustavo De Conti Teixeira Costa, Ph.D

Academic Editor

PLOS ONE

Journal Requirements:

4. Please include a copy of Tables 5, 6, 7, 8, which you refer to in your text on page 17.

Reviewers' comments:

Reviewer's Responses to Questions

**Comments to the Author**

1. Is the manuscript technically sound, and do the data support the conclusions?

Reviewer #1: Yes

Reviewer #2: Partly

2. Has the statistical analysis been performed appropriately and rigorously?

Reviewer #1: Yes

Reviewer #2: Yes

3. Have the authors made all data underlying the findings in their manuscript fully available?

Reviewer #1: No

Reviewer #2: Yes

4. Is the manuscript presented in an intelligible fashion and written in standard English?

Reviewer #1: Yes

Reviewer #2: Yes

Reviewer #1: Reviewer Comments for Manuscript PONED2548435

Title: Assessing the Inter & IntraReliability of a Customised Volleyball Performance Analysis System to Analyse Complexes and the Efficacy of the Associated Skills

General Assessment:

This study presents a thorough investigation into the inter and intrareliability of a bespoke volleyball notational analysis system that incorporates complexes, player position, skill type, and skill efficacy. The research addresses a significant gap in the performance analysis literature b not only assessing reliability for a complex set of variables but also by explicitly monitoring for observer drift over an extended period (nine months). The methodological approach is robust, with a detailed description of the coder training process, appropriate statistical analyses (Cohen's Kappa, Weighted Kappa, ICC), and a substantial dataset. The findings are valuable for researchers and practitioners in volleyball performance analysis, demonstrating that detailed, multifaceted coding systems can achieve high reliability. The manuscript is wellstructured and clearly written. I recommend Minor Revisions prior to publication.

1. Data Availability Statement:

The current statement indicates that data cannot be shared due to an agreement with a professional volleyball organization. While this is understandable, it would be beneficial to clarify if there are any conditions under which the data could be made available (e.g., upon reasonable request to the corresponding author and with permission from the organization) or if a deidentified/minimal dataset could be provided. As per PLOS ONE policy, simply stating data is available on request is insufficient, but a more detailed justification for restricted access is needed.

2. Discussion of Variable Reliability:

The results show considerable variability in Kappa values for certain variables, most notably "Type of Serve" in the intrareliability analysis (κ = 0.48 – 0.96 in Table 4a). While the overall values are acceptable, the discussion would be strengthened by briefly hypothesizing reasons for this high variability in specific matches or blocks. Was it due to a particular match context, a temporary lapse in applying the operational definition, or inherent subjectivity in classifying certain serve types?

3. Limitation of a Single Secondary Coder:

The use of one secondary coder for establishing interreliability is a methodological limitation, albeit a common and often practical one. While the authors justify this within their context, the limitation section should more explicitly acknowledge how this choice might affect the generalizability of the interreliability findings compared to studies using multiple coders.

4. Enhancing Reproducibility:

The operational definitions provided in Tables 1 and 2 are excellent. To further enhance the reproducibility and practical application of this work, the authors could consider including the complete coding template or a screenshot of the Nacsport window as Supporting Information (S1 File).

Reviewer #2: Dear Authors,

I have carefully reviewed your work and appreciate your efforts to contribute to this field of research. While your research topic is of interest and has potential, I regret to inform you that I’m unable to propose acceptance of your manuscript in its current form. As the efficacy scale of volleyball skills was a basic instrument for the research, several methodological shortcomings and limitations were identified that require substantial revisions to ensure the paper's scientific validity.I believe that with careful attention to these areas, your manuscript has the potential to make a valuable contribution to the field. I would like to encourage you to revise your work and resubmit it for consideration.

**Do you want your identity to be public for this peer review?** For information about this choice, including consent withdrawal, please see our Privacy Policy

Reviewer #1: No

Reviewer #2: No

---

## [Author Response · Author response to Decision Letter 1]

22 Oct 2025

To whomever this may concern,

We would like to thank the reviewers for taking the time to read our manuscript, and providing valuable feedback. We have carefully reviewed our manuscript in light of your comments and addressed any points where appropriate. We have provided a more detailed response to each point with the documents provided.

Best wishes,

Steven Nicklin

Corresponding author

---

## [Decision Letter · Decision Letter 1]

12 Nov 2025

Assessing the Inter- & Intra-Reliability of a Customised Volleyball Performance Analysis System to Analyse Complexes and the Efficacy of the Associated Skills

PONE-D-25-48435R1

Dear Dr. Nicklin,

We’re pleased to inform you that your manuscript has been judged scientifically suitable for publication and will be formally accepted for publication once it meets all outstanding technical requirements.

Kind regards,

Gustavo De Conti Teixeira Costa, Ph.D

Academic Editor

PLOS ONE

Additional Editor Comments (optional):

Dear authors, after peer review, I believe the manuscript is of sufficient quality for acceptance into the journal. I congratulate you on your manuscript and encourage you to submit further manuscripts to this journal. Sincerely,

Gustavo De Conti

Reviewers' comments:

Reviewer's Responses to Questions

**Comments to the Author**

Reviewer #1: All comments have been addressed

Reviewer #2: All comments have been addressed

2. Is the manuscript technically sound, and do the data support the conclusions?

Reviewer #1: Yes

Reviewer #2: Yes

3. Has the statistical analysis been performed appropriately and rigorously?

Reviewer #1: Yes

Reviewer #2: Yes

4. Have the authors made all data underlying the findings in their manuscript fully available?

Reviewer #1: Yes

Reviewer #2: Yes

5. Is the manuscript presented in an intelligible fashion and written in standard English?

Reviewer #1: Yes

Reviewer #2: Yes

Reviewer #1: The author has made revisions. The revisions are in line with the suggestions we put forward and respond to them appropriately, meeting the requirements for publication.

Reviewer #2: I've read the authors' updated manuscript and their thorough "Response to Reviewers."

The authors have addressed the issues that have been raised in my initial review in a comprehensive and acceptable manner. They appropriately explained that the goal of this study was not to create and validate a new system, but rather to evaluate the intra- and inter-reliability of their current custom system. Additionally, they pointed out that their scales are based on previously used structures in volleyball literature, which offers crucial background information. By elucidating that "position" refers to "player position," that "efficacy" is an observed 5-point ordinal scale, and by offering the operational definitions for "skill types" as an additional file, they have allayed my worries regarding missing definitions. They also claim to have made a few small clarifications to Table 2's Block definitions.

The authors have taken into account the criticism of the scales' construct validity (such as splitting serve receive levels) by including remarks in the discussion and conclusion that recognize these shortcomings and suggest them as particular directions for further study. Additionally, the authors' own data demonstrated that the variables with the lowest inter-rater reliability scores were, in fact, the ones that I found to be the most subjective (such as "Type of Set" and "Type of Spike"). The authors have improved their paper by addressing this in their discussion.

My concerns have been satisfactorily addressed by the authors. In this version the manuscript is much clearer, its contribution to the field is clearly defined, and its limitations are appropriately contextualized. I recommend acceptance.

**Do you want your identity to be public for this peer review?** For information about this choice, including consent withdrawal, please see our Privacy Policy

Reviewer #1: No

Reviewer #2: **Yes: ** SOTIRIOS DRIKOS

---

## [Editor Report · Acceptance letter]

PONE-D-25-48435R1

PLOS ONE

Dear Dr. Nicklin,

I'm pleased to inform you that your manuscript has been deemed suitable for publication in PLOS ONE. Congratulations! Your manuscript is now being handed over to our production team.

Kind regards,

on behalf of

Dr. Gustavo De Conti Teixeira Costa

Academic Editor

PLOS ONE